# An Emerging Role for Sigma-1 Receptors in the Treatment of Developmental and Epileptic Encephalopathies

**DOI:** 10.3390/ijms22168416

**Published:** 2021-08-05

**Authors:** Parthena Martin, Thadd Reeder, Jo Sourbron, Peter A. M. de Witte, Arnold R. Gammaitoni, Bradley S. Galer

**Affiliations:** 1Zogenix, Inc., Emeryville, CA 94608, USA; pmartin@zogenix.com (P.M.); treeder@zogenix.com (T.R.); agammaitoni@zogenix.com (A.R.G.); 2University Hospital KU Leuven, 3000 Leuven, Belgium; j.sourbron@gmail.com; 3Laboratory for Molecular Biodiscovery, Department of Pharmaceutical and Pharmacological Sciences at KU Leuven, 3000 Leuven, Belgium; peter.dewitte@kuleuven.be

**Keywords:** developmental and epileptic encephalopathy, fenfluramine, sigma-1 receptor, serotonin receptor

## Abstract

Developmental and epileptic encephalopathies (DEEs) are complex conditions characterized primarily by seizures associated with neurodevelopmental and motor deficits. Recent evidence supports sigma-1 receptor modulation in both neuroprotection and antiseizure activity, suggesting that sigma-1 receptors may play a role in the pathogenesis of DEEs, and that targeting this receptor has the potential to positively impact both seizures and non-seizure outcomes in these disorders. Recent studies have demonstrated that the antiseizure medication fenfluramine, a serotonin-releasing drug that also acts as a positive modulator of sigma-1 receptors, reduces seizures and improves everyday executive functions (behavior, emotions, cognition) in patients with Dravet syndrome and Lennox-Gastaut syndrome. Here, we review the evidence for sigma-1 activity in reducing seizure frequency and promoting neuroprotection in the context of DEE pathophysiology and clinical presentation, using fenfluramine as a case example. Challenges and opportunities for future research include developing appropriate models for evaluating sigma-1 receptors in these syndromic epileptic conditions with multisystem involvement and complex clinical presentation.

## 1. Introduction

Developmental and epileptic encephalopathies (DEEs) are debilitating conditions characterized by recurrent pharmacoresistant seizures of multiple types, most notably convulsive seizures, and associated with increased risk of sudden unexpected death in epilepsy (SUDEP), particularly in patients with a high burden of generalized tonic-clonic seizures (GTCs) [1]. Patients with DEEs also present with developmental delay; profound cognitive impairment; impairments in motor function, speech, and language; and growth abnormalities [1,2,3]. The drug fenfluramine, originally described only in terms of serotonin (5-hydroxytryptamine, 5-HT) pharmacology [4,5,6], recently has demonstrated efficacy in treating DEEs in multiple phase 3 trials [7,8,9,10] (Appendix A [7,8,9,10,11,12,13,14,15,16,17]). The profound effects of fenfluramine on seizures as demonstrated in clinical trials cannot be explained by serotonergic activity alone, as other serotonergic drugs have only marginal or inconsistent effects on seizure frequency [18,19,20,21].

In addition to the activity of fenfluramine and its major metabolite norfenfluramine at serotonergic receptors, recent preclinical studies in mouse and zebrafish models have identified previously undocumented positive modulatory activity of fenfluramine at sigma-1 receptors (Sigma1Rs) [22,23] (Appendix A; [4,5,6,22,23,24,25,26,27,28]). This interaction with Sigma1Rs suggests a new mechanism of action for fenfluramine’s antiseizure activity (Table 1 [23,25,26,28,29,30,31,32,33,34,35,36,37,38] and Table 2 [23,39,40,41,42,43,44,45,46,47,48]). Sigma1Rs have long been associated with seizure-related conditions—from amphetamine-induced seizures to epileptic encephalopathies [49,50,51]—but one of the newer, intriguing possibilities of Sigma1R functionality may involve mediation of non-seizure comorbidities associated with DEE pathophysiology. The observation that administration of fenfluramine remediated cognitive deficits in mouse models via Sigma1R activity paralleled clinical findings showing that fenfluramine treatment resulted in clinically meaningful improvements in executive function over time in children and young adults with Dravet syndrome or Lennox-Gastaut syndrome [12,14]. Taken together, these findings have led to further investigation into potential Sigma1R activity in other aspects of DEE neuropathophysiology.

Dravet syndrome is considered a prototypical DEE. Its genetic etiology—loss-of-function mutations in the α_1_ subunit of the NaV_1.1_ sodium channel—has led to the development of animal models with clear genotype-phenotype correlations with clinical pathology. Loss of function of the NaV_1.1_ channel preferentially affects inhibitory gamma aminobutyric acid (GABA)ergic interneurons of the hippocampus, leading to loss of inhibitory input on excitatory glutamatergic neurons, hyperexcitability, and seizures (Figure 1A). Under normal physiological conditions, GABAergic interneurons maintain homeostasis by keeping glutamatergic activity in balance. Loss of NaV_1.1_ activity upsets this balance in favor of excess excitatory and seizure activity (Figure 1A). Clinical and preclinical data show that non-seizure comorbidities of the *SCN1A* mutation occur independently of seizures, although the occurrence of seizures can also result in comorbidities [52]. Many of the clinical comorbidities of Dravet syndrome—for example, neurodegeneration and deficits in motor, cognitive, and behavioral functions—are the result of disturbances in cellular processes. For example, loss of function in GABAergic Purkinje neurons of the cerebellum leads to abnormalities in gait that are independent of seizure activity, and the motor phenotype in animal models recapitulates ataxic clinical observations in humans [35]. The Sigma1R plays a role in reversing these disturbances, suggesting that Sigma1R may be a promising pharmacological target with disease-modifying potential [50,53]. The many functions of Sigma1R that are related to the pathology of DEEs and to central nervous system (CNS) physiology suggest a potential role for Sigma1R in the DEE pathophysiology of seizures and in aspects of higher cognitive function. For example, ANAVEX 2-73 (blarcamesine), which has agonist activity both at the Sigma1R and at muscarinic receptors, is currently in clinical trials for neurodegenerative diseases (NCT03774459, NCT03790709; Alzheimer’s disease, Parkinson’s disease dementia, and Rett syndrome (which manifests with seizures in most patients)) and was granted FDA’s Orphan Drug Designation for infantile spasms in 2016 [41,42,54,55,56,57].

These recent studies (Appendix A) have clarified that fenfluramine possesses dual activities: as a serotonergic agent, acting as a potent 5-HT releaser with agonist activity at 5-HT_1D_, _2A_, and _2C_ receptors, and as a positive modulator of Sigma1R. A working model that explains how these dual activities of fenfluramine restore the balance of inhibitory and excitatory activity is presented in Figure 1B: (1) activity at the 5-HT receptors primarily increases GABAergic signaling, while (2) activity at the Sigma1R primarily dampens glutamatergic hyperexcitability.

This review aims to describe the state of the science and to propose future directions for exploring the role of Sigma1R in modulating antiseizure activity and non-seizure neurodevelopmental deficits for patients with DEEs. Sigma1R may offer promise as a disease-modifying pharmacological target, resulting in optimized treatment strategies for seizures and for non-seizure comorbidities associated with DEEs by restoring the balance of inhibitory (GABAergic) input and excitatory (glutamatergic) output in the brain (Figure 1B).

## 2. Function of Sigma1R

Sigma1R was originally described by Martin and colleagues in 1976 as a subtype of opioid receptor localized to the plasma membrane [58]. Sigma1R was first cloned by Hanner et al. in 1996 [59]. Its pharmacology has been described in detail, most recently in comprehensive reviews [49,51,60]. Sigma1R is an ~25 kDa chaperone protein of 223 amino acids. The crystal structure of the Sigma1R identified by Kruse and colleagues in 2016 revealed one transmembrane domain for each monomer. Sigma1R is an endoplasmic reticulum (ER) chaperone protein localized to the mitochondria-associated membranes (MAMs) [50,51,60,61]. Based on the available evidence, it appears that the diverse functions of Sigma1R are directed toward maintaining calcium homeostasis in the cell. Sigma1Rs act at inositol triphosphate receptors (IP_3_Rs) at the ER-MAM interface to facilitate calcium signaling from ER to mitochondria (Figure 2). Under normal physiological conditions, Sigma1R forms a complex with immunoglobulin-binding protein (BiP) and remains in the inactive state. After activation under conditions of physiological stress (e.g., seizures), Sigma1R dissociates from BiP and translocates to the plasma membrane or the plasmalemmal region to interact with client proteins and elicit physiological responses. Intracellularly, Sigma1R mobilizes calcium stores and modulates neurotransmitters, trophic factors, and ion channels at the level of the plasma membrane [49,62,63,64,65] It is important to note that for consideration of epilepsy, the Sigma1R was described as binding to the NMDA receptor and the G-protein−coupled receptors among other client proteins, which is of critical importance to epilepsy and seizures (Figure 2). Fenfluramine interaction with 5-HT_2A_ and 5-HT_2C_ receptors, Sigma1R, and NMDAR forms a signaling unit that modulates calcium flux through NMDAR. Further, 5-HT_2A_ and 5-HT_2C_ receptors are Gq-coupled receptors, which mobilize calcium influx into the mitochondria through phospholipase C (PLC) and IP_3_ signaling [50]. Coordinated interaction of client proteins (5-HT receptors, NMDAR) and regulatory proteins (histidine triad nucleotide binding protein 1/HINT1, BiP) is hypothesized to mediate antiseizure activity of fenfluramine at the level of the plasma membrane and calcium flux from the cytoplasm into the mitochondrion. Such coordinated interactions mediate the physiological response of Sigma1R modulators on NMDAR-mediated calcium flux in response to cellular stress induced by seizures [24]. The pharmacological effects of Sigma1R ligands are conditioned by signaling proteins, which interact with the receptor, as demonstrated by a recent in vitro study with recombinant proteins [65]. In this report, Sigma1R ligands differentially altered the association of Sigma1R with calcium channels (both NMDAR and transient receptor potential/TRP) and BiP, whereby Sigma1R ligands exhibited biased activity to either promote or disrupt Sigma1R interaction with client proteins in a calcium-dependent manner [65].

In the brain, Sigma1Rs modulate cellular responses to fine-tune neuronal networks and maintain balance between excitatory and inhibitory circuits at the level of neurons and glial cells [66]. Sigma1Rs become activated in response to physiological stressors, including the excitatory/inhibitory neuronal imbalances that lead to seizure activity. Sigma1Rs also associate with Rac1-GTPases (hydrolase enzymes of guanosine triphosphate) in dendrites to modulate redox processes in the formation of dendritic spines [60]. As such, Sigma1Rs have been widely investigated as potentially novel targets for treating neurological disorders, including seizure-related disorders [40].

Sigma1R binds diverse classes of pharmacological ligands [51]. Some of the major high-affinity Sigma1R ligands function as pharmacological agonists or antagonists to result in a physiological response, whereas modulatory ligands have no pharmacological activity by themselves but exert positive or negative modulatory effects when certain physiological processes are activated, or synergistically affect the activity of subtherapeutic concentrations of Sigma1R agonists or antagonists [51,67].

## 3. A Role for Sigma1R in Seizures and Epileptogenesis and GABAergic Signaling

Sigma1R function is required to maintain normal levels of neuronal excitability. Mice with a genetic deletion of Sigma1R (*Sigma1R−/−*) have increased susceptibility to convulsive seizures caused by administration of the chemoconvulsant γ-aminobutyric acid (GABA)_A_ antagonists pentylenetetrazol (PTZ) and (+)-bicuculline (BIC) [39] (Table 2). *Sigma1R−/−* mice also show decreased expression of GABA_B_ R2 subunits, GABA_A_ γ2, both of which lower the seizure threshold [39]. Fenfluramine has been shown to reduce tonic seizure frequency in Lennox-Gastaut syndrome [7], and Vavers et al. (2021) [39] hypothesized that a Sigma1R-GABA_B_ mechanism may play a role in the epileptogenesis of DEEs with a predominantly tonic component [39].

Epileptic seizures result from an imbalance in neuronal excitatory (glutamatergic) and inhibitory (e.g., GABAergic) input. In epilepsy, insufficient GABA_A_ receptor control of glutamatergic signaling, including *N*-methyl-d-aspartate receptors (NMDARs), results in excessive excitatory stimulus and increased seizure activity, which in turn activates Sigma1R signaling responses. Numerous psychotropic drugs bind to Sigma1R, making Sigma1R a potential druggable target for controlling seizures and non-seizure CNS comorbidities related to both seizure activity over time and the underlying pathology of Dravet syndrome and other DEEs [50,52,68].

Dravet syndrome has a well-defined *SCN1A* genetic etiology and well-defined genotype-phenotype correlations in animal models [68]. Loss-of-function mutations in the α_1_ subunit of the NaV_1.1_ sodium channel of GABAergic interneurons in different regions of the brain result in insufficient GABAergic inhibition and increased excitatory glutamatergic activity by multiple mechanisms (Table 1 and Table 2), many of which are also under Sigma1R control (Table 3; [41,49,57,64,69,70,71,72,73,74,75]). Loss of GABAergic inhibition in specific regions of the brain is a key mechanism leading to seizures and non-seizure comorbidities associated with clinical manifestations of Dravet syndrome. Studies in *Scn1a−/−* mouse models of Dravet syndrome have sought to identify neurological sequelae and symptoms of the defective *Scn1a* in mammals (Table 1). In *Scn1a−/−* mice, thermally evoked seizures start in the hippocampus as a result of reduced sodium current in GABAergic inhibitory interneurons that regulate excitatory glutamatergic pyramidal neurons, without affecting sodium currents in the excitatory neurons themselves [29]. Using CRE-Lox conditional knockout technology to target genetic deletion of *Scn1a* to specific brain regions, Catterall and colleagues isolated regions of the brain where loss of GABAergic tone resulted in seizures versus other DEE-related non-seizure clinical manifestations (Table 1). Conditional knockout of *Scn1a−/−* in the hippocampus resulted in thermally induced seizures and learning and memory deficits without the behavioral or motor abnormalities characteristic of *Scn1a−/−* mice [29,32]. In contrast, *Scn1a+/−* targeted to forebrain GABAergic neurons (*Dlx1/2-Scn1a+/−*) led to behavioral abnormalities, including autistic-like social interaction deficits [33,34]. Isolated cerebellar Purkinje neurons from *Scn1a +/−* and *Scn1a −/−* mice (with an ataxic phenotype similar to Dravet cases) showed substantial reductions in the sodium current [35], and GABAergic neurons isolated from the reticular thalamic nucleus disrupted sleep architecture (also similar to Dravet cases) [37]. With the exception of sleep architecture and SUDEP [38], all of these clinical phenotypes have a potential Sigma1R mechanism of action similar to that described in other disease state models (Table 1), and many of these clinical manifestations are responsive to fenfluramine (Appendix A). Taken together, these observations raise the intriguing possibility that fenfluramine’s Sigma1R pharmacology may be acting in concert with its serotonergic activity to ameliorate not only seizures but also non-seizure comorbidities.

## 4. Fenfluramine: From Serotonergic Uptake Inhibitor to Sigma1R Positive Modulator

Fenfluramine has demonstrated profound efficacy for the DEE Dravet syndrome in three phase 3 trials (Appendix A) [8,9,11,76]. Fenfluramine has shown durable suppression of seizure activity for over 3 decades in phase 2 studies [77], and most recently in drug development open-label extension studies of Dravet syndrome, with treatment provided over 3 years for some patients [11]. A more recent phase 3 clinical trial has demonstrated efficacy in Lennox-Gastaut syndrome [7]—a DEE with a more heterogeneous presentation and etiology than Dravet syndrome [78]. Efficacy was most pronounced in patients who presented with GTCs at baseline [7]. Additionally, longer-term seizure control was observed in a recently published phase 2 study [79]. Investigator-initiated studies have demonstrated clinical efficacy of fenfluramine in controlling seizures in other DEEs as well, including CDKL5-associated disorder (CDD) and Sunflower syndrome [15,16,17]. It is important to note that fenfluramine shows evidence of improvement in executive function—a non-seizure-related outcome—after administration over 14 weeks (short term) in Lennox-Gastaut syndrome and Dravet syndrome, and over 1 year (long term) in Dravet syndrome [7,14]. In Sunflower syndrome, the non-seizure-related outcomes of IQ scores and electroencephalographic (EEG) patterns showed improvement, with a trend toward increased mean full-scale IQ scores (*p* = 0.06) and, in several patients, reduction in epileptiform activity and resolution of photo-paroxysmal response observed on EEG after treatment with fenfluramine [17].

Selective serotonin reuptake inhibitors (SSRIs) and serotonin-norepinephrine reuptake inhibitors (SNRIs) are used to treat depression, and SSRIs and selective 5-HT (serotonin) agonists have been evaluated in DEEs. Although some antiseizure activity has been noted, results have been equivocal or marginal [21,80,81], showing that serotonergic activity alone cannot fully account for the effectiveness of fenfluramine.

### 4.1. In Vitro and Ex Vivo Binding and Functional Assays Demonstrate Fenfluramine Is a Positive Modulator of Sigma1R

Early studies demonstrated that fenfluramine bound Sigma1R in crude synaptic membrane preparations from rat brain [27] and in guinea pig brain [22]. Later studies used two in vitro assays and one ex vivo activity assay to demonstrate that fenfluramine binding resulted in positive modulatory activity at the Sigma1R [22]. In the first in vitro activity assay, a well-validated cell-based assay for sigma-1 receptor activation was utilized. This in vitro assay measures the dissociation of Sigma1R from the endoplasmic reticulum stress protein BiP, as quantified by immunoprecipitation. By itself, fenfluramine did not affect the BiP/Sigma1R association. However, in the presence of the sigma-1R agonist PRE-084, low micromolar concentrations of fenfluramine significantly potentiated the activity of PRE-084 in the BiP/Sigma1R dissociation assay. The increase in activity of PRE-084 was observed at all concentrations of PRE-084 tested. In the second in vitro activity assay, fenfluramine and its major metabolite norfenfluramine disrupted the association between Sigma1R and the NR1 regulatory subunit of NMDARs [24]. The positive modulatory activity of fenfluramine was confirmed in the ex vivo vas deferens contraction model of Sigma1R activity. Fenfluramine alone did not induce vas deferens contraction at concentrations up to 10 μM. However, fenfluramine potentiated contractions induced by the Sigma1R agonist (+)-SKF-10,047 [22]. Taken together, results of these experiments support positive modulatory activity of fenfluramine at Sigma1R in vitro and ex vivo.

In addition to the evidence for positive modulation of Sigma1R activity by fenfluramine, the voltage-gated sodium channel antagonist phenytoin, used clinically for more than 80 years to treat seizures, was historically the first Sigma1R positive allosteric modulator to be identified [51,82]. Using crude synaptic guinea pig membrane preparations, phenytoin potentiated binding of the tritiated Sigma1R agonist 3-PPP to Sigma1R, but had no effect on Sigma1R binding to the tritiated Sigma1R antagonist NE-100 (Table 1) [44,45,46,51].

### 4.2. In Vivo Functional Assay Demonstrates Fenfluramine Is a Positive Modulator of Sigma1R

In vivo data further corroborate fenfluramine as a positive modulator of Sigma1R (Appendix A). Fenfluramine was tested in the dizocilpine-induced amnesia model of learning and memory using two well-described behavioral tests validated for evaluating Sigma1R activity in vivo (spontaneous alternation and step-through passive avoidance) [83,84,85,86,87]. Fenfluramine at 0.3 and 1.0 mg/kg significantly enhanced activity of the Sigma1R agonist PRE-084 in attenuating dizocilpine-induced deficits in both behavioral tests. All combination effects of fenfluramine and PRE-084 in both tests were fully blocked by the Sigma1R antagonist NE-100 [22], confirming the behavioral benefits of fenfluramine were mediated by Sigma1R.

### 4.3. Positive Modulators of Sigma1R Have Demonstrated Efficacy across a Spectrum of In Vivo Seizure Models

Positive modulators of Sigma1R have demonstrated antiseizure activity in five different animal models of epilepsy. These models range from acute chemoconvulsant challenge to a model of a developmental epileptic encephalopathy. The types of seizures that were suppressed include tonic, clonic, convulsive, generalized tonic, and generalized tonic clonic seizures, as well as death.

The Sigma1R positive allosteric modulators SKF83959 and SOMCL-668 were tested in three different models of epilepsy [43]. The first model was the maximal electroshock (MES) model in mice. The threshold for eliciting GTCs was assessed. Drugs that are effective against GTCs induced by electroshock are often effective against partial and tonic-clonic seizures in humans. Administration of SKF83959 or SOMCL-668 before MES testing significantly (*p* < 0.05) elevated the threshold to GTCs.

In the second epilepsy model, SKF83959 and SOMCL-668 were tested in the mouse PTZ-induced seizure model. The PTZ-induced seizure is a widely used model for evaluating antiseizure drugs. Drugs that inhibit PTZ-induced seizures have been found to be effective against clonic seizures in humans. Administration of SKF83959 or SOMCL-668 prior to PTZ challenge significantly (*p* < 0.05) prolonged latencies to clonic seizures and GTCs, lowered seizure scores, improved survival time, and lowered mortality [43].

In the third epilepsy model, SKF83959 and SOMCL-668 were tested in a mouse status epilepticus (SE) model, where SE was induced by intraperitoneal injection of kainic acid. Status epilepticus often requires intensive care in clinical practice and is often resistant to traditional antiseizure drugs. As expected, this model was resistant to the commonly used anti-epileptic drug valproate (VPA). In contrast to results with VPA, administration of SKF83959 or SOMCL-668 significantly (*p* < 0.05) prolonged latency to seizures, shortened the duration of seizures, lowered the average severity of seizures, and decreased mortality [43].

To confirm that the activity of SKF83959 and SOMCL-668 in the above models was mediated through the Sigma1R, antiseizure activities of SKF83959 and SOMCL-668 were assessed in the presence of the Sigma1R antagonist BD1047. In the presence of the Sigma1R antagonist, both SKF83959 and SOMCL-668 lost their efficacy in all three of these seizure models, confirming that their antiseizure activity was mediated through interaction with Sigma1R [43].

The Sigma1R positive allosteric modulator E1R was tested for efficacy against clonic and tonic seizures caused by administration of the GABA antagonists PTZ and bicuculline (BIC). Administration of E1R before PTZ challenge significantly (*p* < 0.05) increased the thresholds to PTZ-induced tonic and clonic seizures in a dose-dependent manner. Similarly, administration of E1R before BIC challenge significantly (*p* < 0.05) increased thresholds to BIC-induced tonic and clonic seizures. To confirm that the activity of E1R was mediated through the Sigma1R, the antiseizure activity of E1R was assessed in the presence of the Sigma1R antagonist NE-100. In the presence of the Sigma1R antagonist, E1R lost its efficacy in the PTZ-induced seizure model, which showed that the antiseizure effect of E1R was mediated through Sigma1R activity [40].

Previous work [23,88] in a zebrafish model of the developmental epileptic encephalopathy Dravet syndrome demonstrated that fenfluramine blocks epileptiform activity and hyperlocomotion in the *scn1Lab−/−* mutant zebrafish. In this model, fenfluramine was found to act via a combination of 5-HT_1D_, 5-HT_2A_, 5-HT_2C_, and Sigma1R [23]. Further work tested whether the Sigma1R positive allosteric modulator SOMCL-668 could also affect epileptiform activity and hyperlocomotion in the *scn1Lab−/−* mutant zebrafish. Wild-type and homozygous *scn1Lab−/−* mutant zebrafish larvae were treated with vehicle or test articles at 6 days post-fertilization for a 22-h period [23,30]. Locomotor activity and forebrain local field potentials (LFPs) used to define seizures were measured on post-fertilization day 7. Test articles were the Sigma1R positive modulator SOMCL-668 and fenfluramine. Compared to vehicle-treated controls, SOMCL-668 reduced the hyperlocomotor phenotype of the *scn1Lab−/−* mutant zebrafish larvae but did not affect normal levels of locomotion in the wild-type controls (Figure 3A). SOMCL-668 also significantly (*p* < 0.05) reduced the frequency of epileptiform events in forebrain of the *scn1Lab−/−* mutant zebrafish as assessed by LFP recordings (Figure 3B). As expected, based on prior work (Sourbron et al. 2016) [30], fenfluramine also (1) significantly reduced the hyperlocomotion phenotype of the *scn1Lab−/−* mutant zebrafish but did not affect normal levels of locomotion in the wild-type controls (Figure 3A) and (2) significantly reduced the frequency of epileptiform events in the *scn1Lab−/−* mutant zebrafish (Figure 3B).

This work is consistent with the efficacy of fenfluramine in this model generated in part through activity at the Sigma1R, suggesting the role of both serotonin and Sigma1R pathways in mitigating seizure activity [22]. Collectively, this in vivo work demonstrates that Sigma1R positive modulators have significant antiseizure activity in a wide variety of seizure models, including a model of DEE.

Further supporting a role for positive modulation of Sigma1R in antiseizure activity, the positive allosteric modulator phenytoin has demonstrated antiseizure activity in multiple in vivo epilepsy models, including the MES model [47,51] and a rat model of ischemia-induced epilepsy [48,51]. The antiseizure activity of phenytoin is traditionally attributed to inhibition of voltage-gated sodium channels, but a re-evaluation of the antiseizure mechanism of phenytoin is warranted in light of the recent linkages between Sigma1R and seizures. In summary, three different antiseizure medications (phenytoin, fenfluramine, and ANAVEX2-73), all with Sigma1R agonist or positive Sigma1R modulatory activity, have been shown to reduce seizures in vivo across a variety of animal models of epilepsy.

### 4.4. 5-HT Receptors and Sigma1R

A cooperative mechanism between serotonergic activity and Sigma1R has been described for fenfluramine in a mouse model of *N*-methyl-d-aspartate (NMDA)-kindled seizures. In in vitro binding assays and in this model, fenfluramine activity at 5-HT_2A_ and 5-HT_2C_, together with Sigma1R regulatory coupling to the NMDAR NR1 subunit, downregulated NMDAR-mediated excitatory activity [24]. Subsequent calmodulin (CaM) binding inhibited calcium flux to enable negative control of NMDAR function [24], thereby reducing the excitatory activity of glutamatergic neurons and preventing seizures. In the same model, cannabidiol reduced NMDA-mediated seizures and provided neuroprotection after stroke while promoting morphine antinociception by disrupting the regulatory association of Sigma1R with the histidine triad nucleotide binding protein 1 (HINT1) and the NR1 subunit of NMDARs [24,89,90]. Taken together, these studies provide a potential mechanistic basis for Sigma1R activity in modulating seizure activity at the level of NMDAR-mediated hyperexcitability.

Other drugs classified as 5-HT receptor agonists have antiseizure activity, but these drugs have not, to date, demonstrated clinical efficacy comparable to fenfluramine. The potent 5HT_2C_ receptor agonist lorcaserin demonstrated a 47.7% reduction in mean monthly convulsive seizure frequency in a retrospective case series of 35 patients with treatment-resistant epilepsies, including 20 patients with Dravet syndrome [21]. Although fenfluramine dose-dependently potentiated PRE-mediated BiP dissociation from Sigma1R in in vitro binding assays with recombinant proteins, lorcaserin did not synergistically affect the percentage of BiP associated with Sigma1R as the PRE-084 dose increased [22]. It remains to be established whether this difference in pharmacological mechanisms contributes to the differences in clinical effects.

Fluvoxamine, an SSRI, is a Sigma1R agonist with activity at multiple 5-HT receptors. Activity at 5-HT_2A_ and 5-HT_2C_ receptors alleviated allodynia in the partial sciatic nerve ligation model of neuropathic pain [91], and combined Sigma1R and 5-HT_1A_ activity mediated anti-hedonic effects in a mouse model of anxiety and depression [92,93]. Some evidence suggests that fluvoxamine reduces seizure activity in animal models, including PTZ-kindled seizures in mice, but results of clinical studies have been inconclusive [94]. Although 5-HT−mediated mechanisms clearly play a role in modulating the seizure threshold in knockout mice and MES, and in hippocampal kindled and chemoconvulsant rodent models, results for 5-HT−modulating drugs have been inconsistent [19]. For example, anticonvulsant effects of 5-HT_2A_ and 5-HT_2C_ receptors are abolished or become proconvulsant at higher doses with excessive activation [19]. To date, it is unclear why other serotonin-modulating drugs do not show clinical efficacy comparable to fenfluramine for reducing seizure frequency in DEEs, even with SSRIs that also modulate Sigma1Rs (e.g., fluvoxamine). A combination of fenfluramine dosage (discussed in the next section and in Figure 4); pharmacological mechanisms of action at 5-HT_1D_, 5-HT_2A_, and 5-HT_2C_ receptors and Sigma1R; and Sigma1R positive modulation as opposed to Sigma 1R agonism offers a hypothesis for future testing. Differences in epilepsy models, species, and experimental protocols must also be considered.

### 4.5. Sigma1R Ligands: Dose Considerations

A common feature of Sigma1R agonists is that they follow a bell-shaped biphasic dose-response curve (Figure 4) [22,50,95,96,97,98,99]. The principle of hormesis (defined as the “paradoxical beneficial effect seen with low doses and less beneficial effect at higher doses”) applies to the Sigma1R. The mechanisms for the bell-shaped dose-response curve are not well understood, and this has led to contradictory reporting about the agonist or antagonist activity of Sigma1R ligands and awareness that a clear understanding of the dose-response curve is required for identification of a clinically effective dose. Contradictions in the data tend to be resolved when drugs are tested over a wide enough dose range [97]. For example, the apparently counterintuitive observation that the Sigma1R antagonist NE-100 reduced epileptiform activity and hyperlocomotion in the preclinical *scn1Lab−/−* mutant zebrafish model of Dravet syndrome can be explained in terms of this biphasic dose-response behavior [23].

## 5. Epileptogenesis, Neuroactive Steroids, and Sigma1R

Neuroactive steroids, which are endogenous ligands for Sigma1R, modulate synaptic activity at GABA_A_ and glutamatergic synapses, thereby regulating key pathways in epileptogenesis by impacting the balance between excitatory activity and inhibitory activity in the brain [39,100,101]. These neuroactive steroids include dehydroepiandrosterone (DHEA), dehydroepiandrosterone sulfate (DHEAS), pregnenolone sulfate (PREGS), allopregnanolone, and progesterone. Allopregnanolone is a positive allosteric modulator of GABA_A_, and DHEAS is a positive modulator of NMDAR-mediated responses [101,102]. Neuroactive steroids reduced spontaneous seizures and hyperthermia-induced seizures and prolonged survival in an *Scn1a*+/− mouse model of Dravet syndrome [103]. Specifically, treatment of mice with the neuroactive steroid SGE-516, which is a potent positive modulator of both synaptic and extrasynaptic GABA_A_ receptors, had a significant protective effect. Further, spontaneous GTC seizure frequency was significantly reduced among mice pretreated with SGE-516 [104].

Fenfluramine has been found to attenuate dizocilpine-induced amnesia synergistically when administered with the Sigma1R agonist PRE-084, demonstrating that fenfluramine acts as a positive modulator of Sigma1R [22]. The neuroactive steroids DHEAS and PREGS are Sigma1R agonists that have been shown to attenuate dizocilpine-induced learning deficits in a mouse amnesia model of learning and memory [86]. Co-administration of a low dose of fenfluramine with a low dose of each of these neuroactive steroids, respectively, resulted in synergistic attenuation of learning impairment. The effect of fenfluramine was blocked by progesterone, a Sigma1R antagonist [105], showing that fenfluramine acts as a positive modulator at an endogenous ligand of the Sigma1R. Taken together, these results suggest that a positive modulator of Sigma1R can interact with an endogenous Sigma1R agonist to reverse learning and memory impairment, revealing an intriguing new avenue of research for Sigma1R in mediating both seizures and non-seizure learning deficits in models of Dravet syndrome.

## 6. A Role for Sigma1R in Non-Seizure Comorbidities of DEEs

We now consider discrete elements of non-seizure symptoms associated with DEEs, details of the underlying pathophysiology revealed by genetic models, and evidence suggesting that each pathological mechanism may be modifiable by manipulation of Sigma1R pharmacology.

### 6.1. Neuroplasticity and Connectivity: Sigma1Rs in Maintaining Dendritic Arborization

In addition to mitigating seizure activity, Sigma1Rs regulate neuroplasticity and functional connectivity by coordinating neurogenesis and axon guidance, both of which are highly impaired in preclinical models of Dravet syndrome [66]. In models of Dravet syndrome, abnormal dendritic arborization is a pathological manifestation of *Scn1a−/−* deletion and has been hypothesized to contribute to loss of GABAergic tone in regions of the brain most affected by Dravet syndrome (Table 1). Loss of proper dendritic arborization may also cause behavioral and cognitive deficits by preventing the proper connectivity between neurons and neuronal networks.

In the *Scn1lab−/−^t^* zebrafish model of Dravet syndrome (Appendix A) [28], epileptiform activity and seizures are also accompanied by a significant reduction in the level of dendritic arborization. This morphological pathology was evident before seizure onset, demonstrating that loss of dendritic arborization is not a consequence of seizures and instead may drive epileptogenesis, perhaps by lowering GABAergic connectivity and tone. When larvae were treated with fenfluramine, seizures and epileptiform activity were significantly reduced. Fenfluramine also restored dendritic arborization to wild-type, non-diseased levels. It is important to note that this benefit of fenfluramine is independent of its seizure reduction activity, as treatment with a benzodiazepine (diazepam) also reduced seizures but had no benefit for loss of dendritic arborization (Appendix A) [28]. Prior work has established that Sigma1R function is required for normal levels of dendritic arborization. When Sigma1R is depleted in knockdown cell culture models, axonal growth cones are decreased in size, axonal density, mitochondrial number, and mobility [106]. These changes result in disorganized axonal projections with abnormal circular routes lacking structural organization and architecture, with poor arborization of presynaptic axons, fewer synapses, and aberrant axonal pathfinding [66]. A Sigma1R-dependent mechanism for modulatory activity of fenfluramine in restoring the architecture of dendritic arborization is an intriguing topic to be explored for possible disease-modifying activity in Dravet syndrome and other DEEs.

### 6.2. Neurodegeneration

Neuronal damage to GABAergic arborization includes cortical necrosis and disorderly cellular architecture and produces a phenotype related to the underlying *Scn1a* genetic defect, but seizure-related excitotoxic damage also causes neurodegeneration. Neurodegeneration is evident in *Scn1a−/−* mice as necrosis in cortical regions at 14 days following onset of epileptic activity [31]. Cortical damage was accompanied by apoptosis, mTOR activation indicating metabolic stress, and poor performance on four cognitive function tests evaluating learning and spatial memory [31].

Sigma1Rs play a protective role in many neurodegenerative diseases, including providing a neuroprotective mechanism for reducing excitotoxic damage caused by excessive glutamate in epilepsy [50,69,107] and regulating excitotoxicity caused by glutamatergic hyperactivity associated with seizures and subsequent oxidative stress and reactive oxygen species (ROS). Sigma1Rs are broadly neuroprotective and have been associated with mitigating seizure activity and preventing excitotoxic damage resulting from amphetamine-induced seizures (Table 3) [108,109].

Sigma1R-mediated neuroprotection and antiapoptosis are mechanisms implicated in multiple neurodegenerative disorders such as Alzheimer’s disease, Parkinson’s disease, stroke, and amyotrophic lateral sclerosis (ALS)/frontotemporal dementia [69], as well as in retinal neurodegeneration [110]. Sigma1Rs promote neurogenesis, initiate adaptive neuroplasticity in response to stress, and regulate NMDAR activity, with long-term potentiation in memory formation and neuronal differentiation [49,111]. Cellular stressors disrupt global calcium homeostasis, resulting in Sigma1R translocation, to promote cell survival [50]. Sigma1R protects against apoptosis by increasing expression of antiapoptotic Bcl2, which counteracts ROS/nuclear factor κB (NFκB)-mediated apoptosis [108].

Modulation of Sigma1R pharmacology has been documented to preserve neuronal integrity via multiple pharmacological mechanisms. In Sigma1R knockout mice, hypercytokinemia was induced by acute inflammation through a mechanism involving elevated restricted endonuclease activity of IRE1, an ER stressor [64]. Sigma1R activation and subsequent inhibition of IRE1 signaling by the Sigma1R agonist fluvoxamine reduced inflammation in vivo and slowed inflammatory cytokine release from peripheral blood mononuclear cells [64]. In the brain, similar mechanisms mediate microglial inflammation, and microglial inflammation can be modulated by allosteric modulation of Sigma1R. The potent allosteric modulator Sigma1R SKF83959 inhibited microglial activation and synergistically enhanced the anti-inflammatory effects of the neuroactive steroid DHEA on microglial activation in BV2 microglia [112]. These results suggest a role for Sigma1R modulation in neuroinflammation and microglial hyperactivity, which are components of many neurodegenerative disorders.

Sigma1R coupling with IRE1 may occur as a response to ER stress, which can cause the unfolded protein response and apoptosis signaling in multiple neurodegenerative diseases (e.g., Alzheimer’s, Parkinson’s, Huntington’s, ALS) [113]. Evidence suggesting that Sigma1R activity is required for healthy motor neuron function is especially strong, with mutations in Sigma1R linked to neurodegenerative motor neuron disease in humans (Table 3). A missense mutation in the transmembrane domain of Sigma1R is associated with juvenile ALS, and cells expressing the mutant protein were more prone to ER stress−induced apoptosis [71]. A 3′-UTR (untranslated region) variant was associated with non-juvenile ALS without frontotemporal lobar dementia (FTLD [72]). An in-frame deletion of Sigma1R corresponded with a phenotype of distal hereditary motor neuropathy; functionally, this variant induced ER stress and increased apoptosis [73]. A homozygous missense mutation of Sigma1R was identified in a patient with distal hereditary motor neuropathy and lower limb spasticity (Silver-like syndrome [74]). A non-polymorphic mutation in the 3′-UTR of Sigma1R was identified in FTLD and in motor neuron disease [75]. ALS-linked Sigma1R variants have induced the dissociation of MAM components, disrupted calcium homeostasis via mislocalization of IP3R3, and produced mitochondrial dysfunction and neurodegeneration [70]. In mouse models of ALS, genetic deletion of Sigma1R in the mutant SOD1 was shown to exacerbate the disease. In mouse models, Sigma1R agonists are neuroprotective and have improved motor neuron function in neurodegenerative conditions [50]. The Sigma1R agonist ANAVEX 2-73 (blarcamesine) has demonstrated disease-modifying improvements in preclinical models of the neurodegenerative disorder Rett syndrome, and is currently in phase 2b/3 clinical trials for Alzheimer’s disease (NCT03790709) and Parkinson’s disease dementia (NCT03774459) (Table 3) [41,54,55,56,57]. Together, these data show that Sigma1R activity is required for normal motor neuron function. Motor neuron diseases are linked to glutamate excitotoxicity and dysregulated calcium signaling—activities that Sigma1Rs are known to regulate. Correct Sigma1R function may also be beneficial for reducing the excitotoxic effects of glutamate in chronic neurodegenerative disease (Table 3) [50,69], including the excitotoxic damage notable in epilepsy syndromes.

### 6.3. Cognitive Deficits: Learning and Memory

Virtually all adults with Dravet syndrome exhibit moderate to severe intellectual impairment [114,115,116]. Cognitive deficits are believed to be sequential to prolonged seizure activity and likely occur as a consequence of both underlying genetic pathology and treatment with some antiseizure medications (ASMs) during critical periods of neuronal development [52,117,118]. Cognitive deficits also occur along with seizures in multiple mouse models of Dravet syndrome. For instance, reduced sodium currents in forebrain GABAergic neurons of *Scn1a-/+* mice have led to context-dependent deficits in spatial memory—an effect that was remediated by clonazepam, a positive allosteric modulator of GABA_A_ receptors [33].

Cognitive deficits were originally investigated as a safety endpoint with fenfluramine administration in phase 3 trials of Dravet syndrome and Lennox-Gastaut syndrome to determine whether fenfluramine had negative cognitive impact, which is a side effect of many ASMs [117,118]. An unexpected finding was that treatment appeared to improve aspects of everyday executive function in patients with both Dravet syndrome and Lennox-Gastaut syndrome as measured by scores on the Behavior Rating Index of Executive Function (BRIEF^®^) and BRIEF^®^2 (2nd edition) [8,14]. Improvements were noted at Year 1 of fenfluramine treatment in patients with Dravet syndrome [12]. These findings are concordant with earlier evidence of modest improvements in intellectual functioning after fenfluramine among children with mental retardation or autism [119,120].

The mechanism underlying fenfluramine’s pro-cognitive effects is under investigation. Some of these effects may be explained by serotonergic activity. Fenfluramine interacts with the 5-HT_4_ receptor [26], which has been associated with learning and memory and executive function in humans [121]. However, multiple clinical findings shown in Appendix A raise the intriguing possibility that fenfluramine may positively impact cognition through a Sigma1R-related mechanism. In mice, fenfluramine attenuated dizocilpine-induced learning deficits associated with spatial and working memory, which are models that have been shown to be mediated by Sigma1R. This effect was biphasic and was potentiated by co-administration of the Sigma1R agonist PRE-084, consistent with positive modulatory activity through Sigma1R leading to improved spatial and working memory [22]. It is interesting to note that while the positive modulator SOMCL-668 reduced epileptic activity in zebrafish, the Sigma1R agonist PRE-084 *inhibited* fenfluramine-mediated anti-epileptic activity in zebrafish models. The reason for this discrepancy is unclear but may involve the Sigma1R biphasic, bell-shaped dose-response pattern [50,95]; this discrepancy further highlights the complexity of activity of fenfluramine at Sigma1Rs. Overall, the pharmacology of fenfluramine in complex and multiple mechanisms may work collectively to promote antiseizure activity and positive effects on cognitive function in patients with DEEs, most notably Dravet syndrome and Lennox-Gastaut syndrome.

Sigma1R activity plays a role in cognitive function in other conditions besides DEEs, including schizophrenia and psychotic depression. Fluvoxamine, an SSRI with potent Sigma1R agonist activity, has improved spatial working memory (executive function) in patients with schizophrenia and psychotic depression [122]. In patients with schizophrenia and major depression, aberrant NMDA transmission is a proposed mechanism for cognitive impairment. Postmortem analysis of the brain of schizophrenic patients has showed reduced density of Sigma1R, and positron emission tomography (PET) studies in healthy human volunteers have revealed that fluvoxamine is a Sigma1R agonist [122]. Evidence in animal models supports a role for fluvoxamine in promoting nerve growth factor−induced neurite outgrowth [122]. Fluvoxamine has also improved phencyclidine-induced cognitive deficits independent of serotonergic activity [49]. Further, the Sigma1R agonist igmesine and the neuroactive steroid DHEA have reversed learning deficits, neurobehavioral impairment, and disordered neuronal architectural alteration observed in juvenile rats exposed in utero to cocaine [49]. A role for Sigma1R has also been described in dendritic spine arborization, branching, and motility in neuronal communication and in synaptic activity (spine density, morphology) for mitigating cognitive deficits associated with Parkinson’s, schizophrenia, autism, obsessive-compulsive disorder (OCD), mental retardation, and Alzheimer’s [28,49,66,106]. In summary, cognitive deficits, which occur as part of DEE presentation, may be improved by the Sigma1R mechanisms related to learning and memory and to neuronal architectural organization.

### 6.4. Behavior: Hyperactivity, Social Interaction, and Autistic-like Behavior

Hyperactivity, irritability, autistic symptoms, and psychosis are among the major causes of institutionalization among older adolescents and adults with DEEs, particularly Lennox-Gastaut syndrome [123]. Patients with Dravet syndrome typically present with profound cognitive-behavioral and cognitive deficits, including autistic-like behaviors and OCD [124]. Hyperactivity, attention deficit disorders, and inability to socialize were reported in surveys as behavioral concerns of caregivers [116]. *Scn1a−/−* animal models recapitulate some of the behavioral phenotypes observed in patients with Dravet syndrome. In mouse *Scn1a−/−* models, reduced sodium currents in forebrain GABAergic neurons have led to social interaction deficits and stereotyped behaviors—effects that were remediated by clonazepam, a positive allosteric modulator of GABA_A_ receptors [33].

Sigma1Rs have been implicated in diverse behavior-related effects similar to those observed in patients with DEEs, some of which are modulated by pharmacological activity at Sigma1R. For example, Sigma1R ligands have been implicated in disorders associated with behavioral abnormalities and abnormal social interactions, including neurocognitive adaptations in autism spectrum disorder [49]. Fenfluramine treatment has resulted in modest improvements in cognitive function among patients with autism and attention deficit hyperactivity disorder; specifically, fenfluramine blunted agitation and hyperarousal in children described as presenting with “severe hyperactivity and inattention that manifested both in the classroom and at home” [119,120]. Taken together, these results open the intriguing idea of a possible role for Sigma1R in mediating some of the behavioral effects associated with DEEs.

### 6.5. Emotional Regulation: Anxiety-like Symptoms or Depression-like Symptoms

Patients with DEEs show impairment in regulation of emotions and mood, which is recapitulated in genetic models of Dravet syndrome. *Scn1a+/−* mice exhibit emotional imbalances, including anxiety-related behavior and atypical fear expression [36]. In a small population of patients with Dravet syndrome treated with fenfluramine, the Emotional Regulation Index of the BRIEF^®^2 instrument used to assess cognitive function showed improvement after 1 year. This effect was most pronounced in patients experiencing clinically meaningful (≥50%) or profound (≥75%) reductions in monthly convulsive seizure frequency (Appendix A) [12]. The mechanism for these observed improvements in affective behavior is unclear, but modulation of Sigma1R pharmacology by fenfluramine may contribute.

Sigma1Rs have been associated with control of mood and emotions in psychiatric conditions such as affective disorder. For example, depression and some of the negative symptoms of schizophrenia can be mitigated by Sigma1R modulation, and some antidepressants used clinically have sub-micromolar affinity for the Sigma1R [49]. In general, Sigma1R ligands are anxiolytic and antidepressant, especially when interacting with endogenous neuroactive steroid ligands [50]. Further, Sigma1R knockout mice have exhibited a depression-like phenotype, and Sigma1R agonists (e.g., fluvoxamine) have ameliorated effects of depression and stress-induced cerebral atrophy.

In mice, the synthetic compound SOMCL-668, a highly selective, potent Sigma1R allosteric modulator, has reduced immobility times in the forced swim test and in tail suspension tests (model for depression) and has improved times in the open-field test (model for anxiety). SOMCL-668 has restored brain-derived neurotrophic factor (BDNF) in the hippocampus that was suppressed in a mouse model of chronic stress [125]. Further, emotional disturbances and memory dysfunction associated with chronic osteoarthritic pain were ameliorated in a mouse model by Sigma1R antagonist E-52862. In this study, Sigma1R antagonists reduced the density of Sigma1R-abundant microglial cells in prelimbic and infralimbic areas of the medial prefrontal cortex without affecting functionality [126]. Although depression-like symptoms were diminished with administration of E-52862 in the context of chronic inflammatory pain, anxiety-like responses were not [126]. Neuroactive steroids including DHEA and DHEAS—endogenous ligands for Sigma1R—have relieved symptoms of major depression in humans and have diminished depression-like behaviors in mouse models of depression [49]. Taken together, these results suggest that Sigma1R may offer a plausible mechanism and pharmacological target for improving mood-related symptoms in patients with Dravet syndrome and other DEEs.

## 7. Limitations and Avenues for Future Research

The importance of Sigma1R signaling in multiple neurological diseases, including epilepsies, is now well established. However, Sigma1R signaling is complex, and many aspects are not well understood. Several avenues of future research as described below will allow the research community to better understand the physiological roles this important receptor plays in both health and disease.

### 7.1. The Sigma1R Interactome

Sigma1R can clearly interact with a wide variety of different proteins. One informative avenue for future research involves building a comprehensive list of the Sigma1R interactome components. Once the interactome is understood, it will be important to understand how consistent these protein interactions are across the different cell types that express Sigma1R. It will be equally important to understand how interactome items change in response to external signaling events or changes in the internal state of the cell. Answers to these basic research questions will lead to better understanding of the functional significance of each individual interaction and will reveal whether the different interactions can be classified into a smaller number of fundamental signaling units.

### 7.2. The Distribution of Sigma1R across Cellular Compartments

Similar to the issue with the Sigma1R interactome, Sigma1R clearly has a complex cellular localization profile. One avenue for future research will involve quantitating the distribution of Sigma1R among different cellular compartments and across different cell types. It will be equally important to understand how this cellular distribution changes in response to external signaling events or internal cellular states. Mapping the changes in cellular compartmentalization caused by different signaling events may enhance our understanding of the fundamental role(s) Sigma1R plays in responding to different external stimuli or internal states.

### 7.3. Sigma1R and Epilepsy

The ubiquity of Sigma1R makes it difficult to isolate seizure and non-seizure activities. Research is warranted to develop experimental methods that isolate Sigma1R activity in seizure- and non−seizure-related mechanisms in DEEs as potential disease-modifying therapy. One approach that can isolate the importance of Sigma1R in DEEs would involve developing conditional CRE knockout models of Sigma1R within the context of a genetically based DEE. This would allow researchers to explore the functional role of Sigma1R in the context of different developmental phases and/or within different pathologies of the particular DEE.

### 7.4. The Role of Sigma1R in SUDEP

SUDEP and disrupted sleep architecture are clinical outcomes of Dravet syndrome that have been recapitulated in *Scn1a*-deficient animal models (Table 1) [37,127], but a possible Sigma1R-related pharmacological target remains to be evaluated. The potential for synergy with 5-HT receptors is supported by the literature for both outcomes, raising the intriguing possibility of a Sigma1R connection. In disrupted sleep architecture, 5-HT_2C_ receptors have been implicated in hippocampal theta rhythms in sleep-wake cycles [128], and in patients with depression disorder and disrupted sleep caused by imbalances in neuroactive steroids, wake therapy has corrected some of these imbalances, resulting in improved depressive symptoms and improved sleep quality [129,130]. Patients with Dravet syndrome have a six-fold higher rate of SUDEP than those with generalized epilepsy, and epidemiological evidence suggests that treatment with fenfluramine has resulted in lower SUDEP incidence [13]. Two different lines of evidence in preclinical models have demonstrated prolonged survival with fenfluramine acting at 5-HT receptors or a neuroactive steroid (i.e., an endogenous ligand for Sigma1R, presumably acting, at least in part, by a Sigma1R-mediated mechanism). In the first case, fenfluramine prolonged survival in both DBA-1 and 129/SvTer rodent models of SUDEP. Subsequent experiments in the DBA-1 model identified a centrally mediated mechanism of preventing respiratory arrest by acting at 5-HT_4_ receptors at doses that were ineffective for seizure control in a DBA-1 mouse model of SUDEP [25,26,131,132]. In the second case, the neuroactive steroid SGE-516 prolonged survival in *Scn1a+/−* mice, with 71% of mice surviving to 6 weeks after sGE-516 compared with ~50% lethality at 1 month in untreated animals [104]. Exploring a possible Sigma1R-mediated mechanism for SGE-516−mediated and fenfluramine-mediated survival may reveal a role for synergy between 5-HT receptors and Sigma1R in SUDEP.

## 8. Conclusions

In conclusion, Sigma1R function (or dysfunction) is linked to multiple facets of the DEE phenotype, including both seizures and non-seizure comorbidities. In this context, it is notable that fenfluramine—with profound antiseizure activities in two DEEs (Dravet and Lennox-Gastaut syndromes)—was recently discovered to be a positive modulator of Sigma1R activity, in addition to its known 5-HT activity. The empirical evidence reviewed in this report suggests a model whereby fenfluramine restores the homeostatic balance between inhibitory GABAergic and excitatory glutamatergic activity to dampen seizure activity in Dravet syndrome and other DEEs (Figure 1). Data from zebrafish models suggest that fenfluramine restores dendritic arborization of GABAergic neurons. At glutamatergic synapses, fenfluramine appears to coordinate with endogenous ligands (e.g., neuroactive steroids) to positively modulate Sigma1R-mediated interaction with the NMDAR, thereby dampening calcium flux and reducing seizure activity at glutamatergic synapses. Further, coordinated interaction with Sigma1R and agonism at 5HT_1D_, 5HT_2A_, and 5HT_2C_ receptors are hypothesized to modulate calcium influx into the ER via a Gq/IP_3_R-mediated mechanism (Figure 2). Control of calcium flux via Sigma1R-mediated mechanisms further helps to restore the balance between inhibitory and excitatory input to decrease seizure activity. Further research into the role of Sigma1R in DEE pathology is warranted, including its role in seizure control and in non-seizure outcomes associated with DEE.

## Figures and Tables

**Figure 1 ijms-22-08416-f001:**
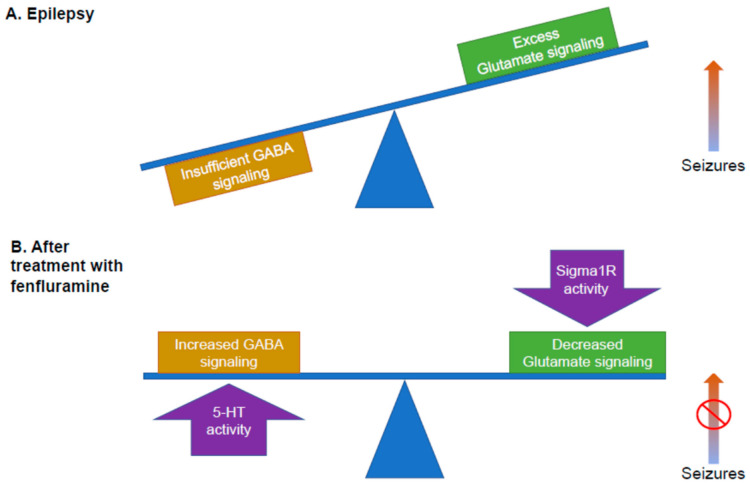
Potential mechanism of fenfluramine. Epilepsy is thought to generate an imbalance between inhibitory GABA signaling and excitatory glutamatergic signaling, leading to seizures. In the above figures, the balance between inhibitory and excitatory signaling is indicated by the blue balance beam. (**A**) Too much glutamate signaling results in seizures (e.g., the right arm of the balance beam moves upward). (**B**) Balance between inhibition and excitation is restored by the dual activities of fenfluramine: the activity of fenfluramine at 5-HT receptors results in increased GABA signaling, boosting inhibition, while the activity of fenfluramine at the Sigma1R reduces glutamatergic signaling, decreasing excitation. This mechanism is consistent with much, but not all, of the published research and is presented as a working model. 5-HT, serotonin; GABA, gamma aminobutyric acid; Sigma1R, Sigma1 receptor.

**Figure 2 ijms-22-08416-f002:**
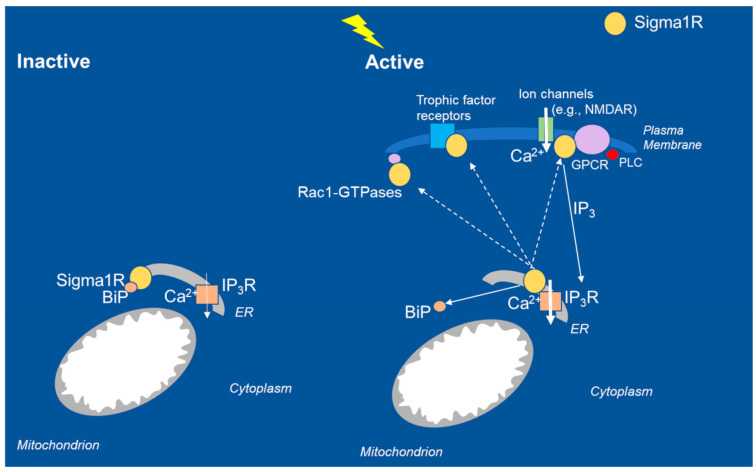
Intracellular interactions of Sigma1R chaperone proteins in inactive and active states. In the resting state, Sigma1R associates with the regulatory protein BiP at the ER MAM. During seizures, loss of GABAergic tone results in excessive NMDAR-mediated calcium influx in glutamatergic neurons. The Sigma1R translocates to the plasma membrane or the ER plasmalemmal space, where it interacts with client proteins, including ion channels (e.g., NMDA, sodium, potassium, voltage-regulated chloride), trophic factor receptors, and kinases; interaction with Rac-GTPases promotes dendritic spine formation and affects neuronal redox processes. For comprehensive consideration of intracellular interactions, see Su 2010, Hayashi and Su 2005, Voronin 2020, Rosseaux and Greene 2015, Vavers 2019, and Maurice 2020. BiP, binding immunoglobulin protein; Ca^2+^, calcium; ER, endoplasmic reticulum; GPCR, G-protein−coupled receptor; GTPase, hydrolase enzyme of guanosine triphosphate; IP_3_R, inositol triphosphate receptor; MAM, mitochondrial associated membrane; NMDAR, *N*-methyl-d aspartate.

**Figure 3 ijms-22-08416-f003:**
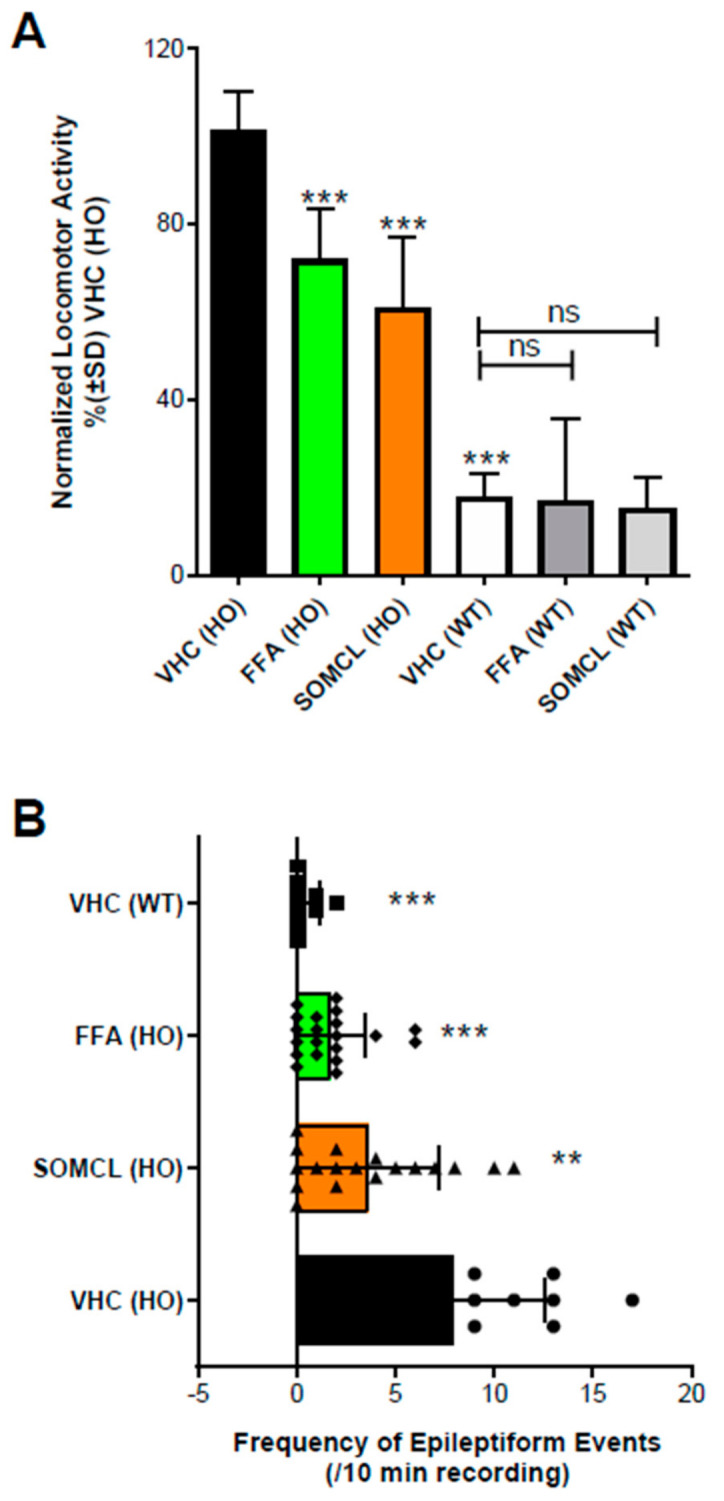
Activity profile of SOMCL-668 (SOMCL, 10 µM) and fenfluramine (FFA, 25 µM) in 7 dpf zebrafish larvae. (**A**) Treatment with SOMCL or FFA decreases the locomotor activity of 7 dpf *scn1Lab−/−* mutants (HO), but not wildtype (WT). Locomotor activity was normalized against VHC-treated *scn1Lab−/−* mutant larvae and is displayed as percentage ± SD. N = 30 to 60 zebrafish larvae for all experimental conditions. (**B**) Treatment with SOMCL and FFA reduces the epileptiform brain activity of 7 dpf *scn1Lab−/−* mutants (HO). The bars represent the numbers of epileptiform events (during 10 min of recording; ± SD) of homozygous *scn1Lab−/−* mutants (HO) or wildtype *scn1Lab+/+* (WT) treated with vehicle (VHC) or SOMCL or FFA, compared to the outcome observed in VHC-treated homozygous *scn1a* mutants, vehicle control (VHC [HO]). N = 9 to 18 zebrafish larvae for all experimental conditions. Overall, statistical significance is represented by asterisks: ** *p* < 0.01, and *** *p* < 0.001 vs. VHC (HO). “No statistical difference” is left blank or is shown by “ns”. Statistical analyses: GraphPad Prism 5 software (GraphPad Software, Inc., San Diego, CA, USA) was used for statistical analyses. (**A**) Locomotor activity was analyzed by one-way ANOVA and subsequent Dunnett’s multiple comparison tests, as described in Sourbron 2016 and 2017 [23,30]. (**B**) Electrographic brain activity was analyzed by Mann-Whitney U tests, as was described in Sourbron 2016 and 2017 [23,30].

**Figure 4 ijms-22-08416-f004:**
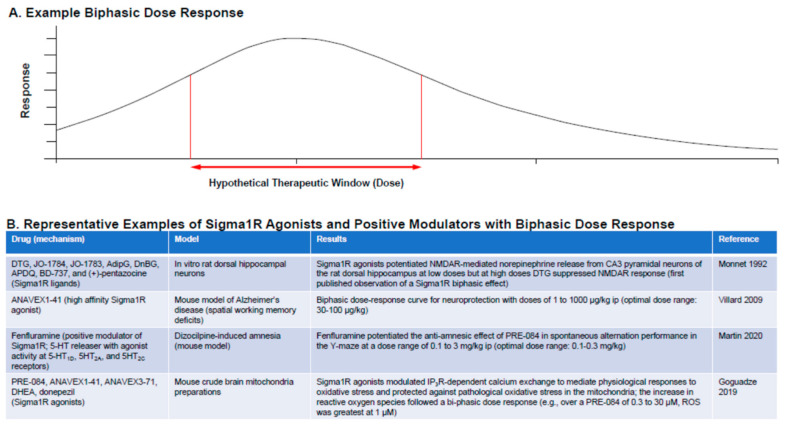
Biphasic dose response is a characteristic of many Sigma1R ligands. (**A**) Example dose–response curve showing therapeutic window for a response. (**B**) Representative examples of Sigma1R agonists and positive modulators following biphasic dose response [22,96,97,98,99].

**Table 1 ijms-22-08416-t001:** Pharmacological targets for clinical phenotypes revealed by genetic models of Dravet syndrome.

Phenotype	DEE Model	Molecular Mechanism	Reference
Seizures	*Scn1a*−/− and +/− mice	Reduced Na currents in GABA interneurons led to hyperexcitability of glutamatergic neurons in the hippocampus	Yu 2006 [29]
	Zebrafish *scn1Lab* model of Dravet syndrome	Epileptiform activity occurred with reduced neural serotonin expression	Sourbron 2016 [30]
	*scn1lab−/−* zebrafish model of Dravet syndrome	GABAergic neuronal loss and astrogliosis (attenuated by fenfluramine) were noted	Tiraboschi 2020 [28]
Neurodegeneration, cognitive deficits	*Scn1a*−/− mice	Cortical necrosis with cognitive deficits was evident in *Scn1a−/−* mice (rescued by liraglutide, a GLP-1 analogue)	Liu 2020 [31]
Cognitive defects, learning and memory	Cre-Lox *Scn1a* deletion localized to hippocampus in mice	Local loss of Nav1.1 function in the hippocampus selectively reduced inhibitory neurotransmission, resulting in thermally evoked seizures and spatial learning deficits	Stein 2019 [32]
	*Scn1a*−/− and +/− mice	Reduced sodium currents in forebrain GABAergic neurons led to context-dependent spatial memory (rescued by clonazepam, a positive allosteric modulator of GABA_A_ receptors)	Han 2017 [33]
Hyperactive behavior	Zebrafish *scn1a* mutant model of Dravet syndrome	Sigma1R agonists abolished FFA-mediated inhibition and hyperlocomotion	Sourbron 2017 [23]
Autistic-like behavior	*Scn1a*−/− and +/− mice	Reduced sodium currents in forebrain GABAergic neurons led to social interaction deficits and stereotyped behaviors (rescued by clonazepam, a positive allosteric modulator of GABA_A_ receptors)	Han 2017 [33]
	*Scn1a*−/− and +/− mice	Reduced sodium currents in forebrain GABAergic neurons led to autistic-like social interaction deficits (rescued by CBD by antagonism at lipid-activated GPR55)	Kaplan Catterall 2017 [34]
Motor: Ataxia	*Scn1a*−/− and +/− mice	Reduced sodium currents in Purkinje cerebellar neurons led to ataxia	Kalume 2007 [35]
Emotional regulation	*Scn1a* +/− mice	Anxiety-related thigmotactic behavior and atypical fear expression were noted	Bahceci 2020 [36]
Disrupted sleep architecture	*Scn1a*−/− and +/− mice	Reduced sodium currents in the GABAergic reticular nucleus of the thalamus resulted in disrupted sleep architecture	Kalume 2015 [37]
SUDEP	*Scn1a* +/− mice and conditional brain- and cardiac-specific KO mice	SUDEP following GTCs in *Scn1a+/−* mice was caused by parasympathetic CNS hyperactivity, leading to lethal bradycardia	Kalume 2013 [38]
	Mouse SUDEP model	Seizure-induced respiratory arrest (attenuated by fluoxetine at 5-HT_3_ without affecting seizures) occurred	Faingold 2016 [25]
	In vivo mouse DBA-1 model of SUDEP	Seizure-induced respiratory arrest (inhibited by FFA activity at 5-HT_4_) occurred	Faingold 2019 [26]

Pharmacological or functional activity has not been reported at 5-HT_3_, 5-HT_5_, 5-HT_6_, or 5-HT_7_ receptors. 5-HT, serotonin; CBD, cannabidiol; CNS, central nervous system; DEE, developmental and epileptic encephalopathy; FFA, fenfluramine; GABA, γ-aminobutyric acid; GLP, glucagon-like peptide; GPR55, G-protein-coupled receptor 55; GTC, generalized tonic-clonic seizure; KO, knockout; Sigma1R, sigma-1 receptor; SUDEP, sudden unexpected death in epilepsy.

**Table 2 ijms-22-08416-t002:** Role of Sigma1R in seizures and epileptogenesis.

Model	Molecular Mechanism	Reference
Dravet zebrafish model	Epileptogenic activity was inhibited by administration of the Sigma1R positive allosteric modulator SOMCL-668	Reported here
Zebrafish *scn1a* mutant model of Dravet syndrome	Sigma1R agonist PRE-084 and 5-HT_1D_ and 5-HT_2C_ antagonists reduced FFA-mediated inhibition of epileptiform activity and completely abolished FFA-mediated hyperlocomotion	Sourbron 2017 [23]
Sigma1R knockout mice	Susceptibility (reduced threshold) to pentylenetetrazol (PTZ) and (+)-bicuculline (BIC) infusion-induced acute seizures was increased	Vavers 2021 [39]
NMDA-kindled seizures in mice	Sigma1R/HINT1 association with NR1 was disrupted by fenfluramine, ameliorating seizures	Rodriguez-Munoz, 2018 [24]
PTZ- and BIC-induced seizure model in mice	Seizure activity was inhibited by administration of the Sigma1R positive allosteric modulator E1R; antiseizure activity of E1R was blocked by the Sigma1R antagonist BD1047	Vavers 2017 [40]
Phase 1 studies of ANAVEX 2-73 (blarcamesine) for infantile spasms (West syndrome)	Sigma1R agonist was granted Orphan Drug Designation for treatment of infantile spasms; preclinical validation for potential treatment for Rett syndrome and other pediatric or infantile disorders with seizure pathology included seizure reduction in rodent chemoconvulsant models and synergistic activity with the antiseizure medications ethosuximide, valproate, and gabapentin	Rebowe 2016 [41]
Maximal electroshock seizure (MES) model in mice PTZ-induced tonic seizure model in mice	Tonic seizure activity was inhibited by administration of ANAVEX2-73, a Sigma1R agonist (pluripotent modulator) with muscarinic activity; administration of subtherapeutic doses of ANAVEX2-73 showed synergistic antiseizure activity in combination with the ASMs ethosuximide, valproate, and gabapentin in the MES model	Rebowe 2015 [42] Rebowe 2015 [42]
SCZ-induced tonic seizure model in mice		Rebowe 2015 [42]
MES model in mice		Guo 2015 [43]
PTZ-induced seizure model in mice Kainate acid−induced status epilepticus model in mice	Seizure activity was inhibited by administration of the Sigma1R positive allosteric modulators SKF83959 and SOMCL-668; antiseizure activity of SKF83959 and SOMCL-668 was blocked by the Sigma1R antagonist BD1047	Guo 2015 [43] Guo 2015 [43]
Rat hippocampal slices (ex vivo)	Epileptiform local field potentials were attenuated by the Sigma 1R positive allosteric modulators SKF83959 and its derivative SOMCL-668	Guo 2015 [43]
In vitro guinea pig brain homogenates	The antiseizure medication phenytoin acted as a potent positive allosteric modulator of Sigma1R activity by augmenting binding of the Sigma1R agonists (+) pentazocine, (+)-SKF-10,047, PRE-084, and (+)-3-(3-hydroxyphenyl)-*N*-(1-propyl) piperidine hydrochloride to solubilized Sigma1R binding sites in homogenized guinea pig brain without affecting binding of the tritiated antagonist NE-100	Chaki 1996 [44] Cobos 2005 [45] DeHaven 1993 [46]
MES model in mice	The positive allosteric Sigma1R modulator and antiseizure medication phenytoin (5−20 mg/kg i.p.) induced potent antiseizure activity via inhibition of voltage-gated sodium channels Phenytoin (20 or 40 mg/kg/day orally) inhibited tonic extension seizures, clonic seizures, and wild running in rats with ED_50_ of 5.0, 10.8, and 20.7 mg/kg, respectively	Jones 1981 [47] Vavers 2019 [51] Edmonds 1996 [48]
Ischemia-induced epilepsy model in rats		

5-HT, serotonin; ASM, antiseizure medication; ED_50_, effective dose in 50% of the study population; FFA, fenfluramine; HINT1, histidine triad nucleotide binding protein 1; i.p., intraperitoneal injection; NMDA, *N*-methyl-d-aspartic acid; PRE-084, Sigma1R agonist; PTZ, pentylenetetrazol; SCZ, semicarbazide; Sigma1R, sigma-1 receptor; SOMCL-668, highly selective potent Sigma1R allosteric modulator.

**Table 3 ijms-22-08416-t003:** Non-seizure DEE-like phenotypes associated with Sigma1R genetic deficiency models or genetic linkage analyses in other disorders.

Phenotype	Model	Molecular Mechanism	Reference
Inflammatory stress	Sigma1R knockout mice	Hypercytokinemia was induced by acute inflammation with restricted endonuclease activity of IRE1, an ER stress sensor	Rosen 2019 [64]
Memory	Sigma1R knockout in APP_Swe_ Alzheimer’s disease mouse model	Memory impairment was exacerbated with increased oxidative stress in the hippocampus	Ryskamp 2019 [69]
	Phase 2a clinical trials (phase 2b/3 ongoing)	ANAVEX 2-73 (blarcamesine), a Sigma1R agonist with some muscarinic modulatory activity, reduced cognitive decline in a longitudinal 148-week study of patients with Alzheimer’s disease	Hampel 2019 [57]; NCT03790709
	Fmr1-KO2 mouse model of fragile X-autism-related disorders model	ANAVEX 2-73 (blarcamesine) reversed Fmr1-KO2 deficit in learning and memory as measured by percent freezing during observation	Rebowe 2016 [41]
Mood	Sigma1R knockout mice	Depression-like phenotype was noted	Hayashi 2011 [49]
Motor, cognitive deficiency	Cellular models of Huntington’s disease	mHtt (huntingtin fragment) protein downregulated Sigma1R expression in cultured PC6.3 neurons	Ryskamp 2019 [69]
Hyperactivity	Fmr1 KO2 mouse model of Fragile X-autism-related disorders model	ANAVEX 2-73 (blarcamesine) reversed Fmr1-KO2 hyperactivity in locomotor assays (number of squares crossed per unit time)	Rebowe 2016 [41]
Motor	SOD1/Sigma1R mouse model of ALS	ALS was exacerbated by disrupting MAM components, dysregulating calcium, and causing mitochondrial dysfunction leading to neuronal degradation	Watanabe 2016 [70]
Motor/Neuroprotection	Genetic linkage analysis and cellular expression of mutant protein	Sigma1R genetic variant was associated with ALS; expression of motor neuron cell line model resulted in loss of resistance to ER stress-induced apoptosis	Al Saif 2011 [71]
Motor/Neuroprotection	Genetic linkage analysis	Sigma1R mutations were associated with non-juvenile ALS without FTLD	Ulla 2015 [72]
Motor/Neuroprotection	Genetic linkage analysis	Sigma1R mutations were associated with nuclear aggregates, ER stress, and increased apoptosis	Li 2015 [73]
Motor	Genetic linkage analysis	Homozygous missense variant of SigmaR1 resulted in distal hereditary motor neuropathy and lower limb spasticity (Silver-like syndrome)	Horga 2016 [74]
Motor	Genetic linkage analysis	Non-polymorphic mutation in Sigma1R was associated with FTLD and ALS	Luty 2010 [75]
Motor	Sigma1R knockout mice	Vulnerability of S1R knockout mice to nigrostriatal axonal degradation was increased in a model of Parkinson’s disease	Ryskamp 2019 [69]

Not an exhaustive list. ALS, amyotrophic lateral sclerosis; DEE, developmental and epileptic encephalopathy; ER, endoplasmic reticulum; FTLD, frontotemporal lobar degeneration; KO, knockout; MAM, mitochondria-associated membrane; Sigma1R, sigma-1 receptor.

## Data Availability

Not applicable.

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
