# Peer review of "An Emerging Role for Sigma-1 Receptors in the Treatment of Developmental and Epileptic Encephalopathies"

_ijms, 2021, doi:10.3390/ijms22168416_

Round 1

Reviewer 1 Report

This is a very comprehensive and interesting review of the possible role of Sigma-1 Receptors in eilepsy in general and epileptic encephalopathies in particular with implications for possible treatment.

Recommendations:

1.  On page 6, the description of the function and localization of the Sigma 1 receptor protein is sufficiently complex, as described below (in ER to mitochondria to plasma membrane  that a cartoon of the protein, it's location in the cell in the active and inactive states, etc. would be helpful to understand the changes it undergoes and effects it has.

"Sigma1R 108 is an endoplasmic reticulum (ER) chaperone protein localized to the mitochondria-associated mem-109 branes (MAMs) , which govern calcium signaling and reactive oxygen species 110 (ROS) homeostasis. Sigma1Rs act at inositol triphosphate receptors (IP3Rs) at the ER-MAM interface to facilitate calcium signaling from ER to mitochondria. Sigma1R forms a complex with 112 immunoglobulin-binding protein (BiP) in the inactive state, dissociates from BiP after activation, and translocates to the plasma membrane to interact with client proteins and elicit physiological 114 responses. Intracellularly, Sigma1R mobilizes calcium stores and modulates neurotransmitters, trophic factors, and ion channels at the level of the plasma membrane. "

2.  page 11.  The discussion of a "bell-shaped, biphasic dose response curve" should be enhanced with a diagram of the dose response curve to give a better sense of how narrow the therapeutic window is before the dose becomes ineffective.  This is germane to comment 3, below

3.  page 14.  The authors discuss the potency of fluvoxamiine as a Sigma 1 R receptor agonist, yet the literature on fluvoxamine's effect on seizures is mixed.  If fluvoxamine is a potent Sigma 1 R receptor agonist, why isn't it as effective for seizures as fenfluramine?  It could be a function of the biphasic dose response curve, but this requires further discussion and elaboration.  

Author Response

Reviewer #1: 

This is a very comprehensive and interesting review of the possible role of Sigma-1 Receptors in eilepsy in general and epileptic encephalopathies in particular with implications for possible treatment. 

Recommendations: 

  1. On page 6, the description of the function and localization of the Sigma 1 receptor protein is sufficiently complex, as described below (in ER to mitochondria to plasma membrane that a cartoon of the protein, it's location in the cell in the active and inactive states, etc. would be helpful to understand the changes it undergoes and effects it has. 

"Sigma1R 108 is an endoplasmic reticulum (ER) chaperone protein localized to the mitochondria-associated mem-109 branes (MAMs) , which govern calcium signaling and reactive oxygen species 110 (ROS) homeostasis. Sigma1Rs act at inositol triphosphate receptors (IP3Rs) at the ER-MAM interface to facilitate calcium signaling from ER to mitochondria. Sigma1R forms a complex with 112 immunoglobulin-binding protein (BiP) in the inactive state, dissociates from BiP after activation, and translocates to the plasma membrane to interact with client proteins and elicit physiological 114 responses. Intracellularly, Sigma1R mobilizes calcium stores and modulates neurotransmitters, trophic factors, and ion channels at the level of the plasma membrane. " 

Response

Please see the new Figure 1 on Page 4, which depicts a conceptual model of the potential mechanism of fenfluramine and the new Figure 2 on Page 8, which depicts active and inactive states after Sigma1R ligand binding, including translocation. We have expanded the text on Pages 3 and 7-8 for clarification. We have rewritten the Conclusions for clarity (see Page 19, Lines 726-736).

  1. page 11. The discussion of a "bell-shaped, biphasic dose response curve" should be enhanced with a diagram of the dose response curve to give a better sense of how narrow the therapeutic window is before the dose becomes ineffective.  This is germane to comment 3, below

 Response

Please see the new Figure 4 on Page 14 (and text on Pages 13-14), which illustrates how biphasic dose responses affect the therapeutic window. In this context, fenfluramine may have a wider therapeutic window than other Sigma1R modulators or greater effectiveness at positive modulation, or the potency or selectivity of combined 5-HT receptor and/or Sigma1R interactions could be a mediator of fenfluramine pharmacology in evoking its antiseizure effects (also discussed in (3) below). The data described in Maurice 2020 illustrate the biphasic dose response. We have referenced Maurice et al. (2020) in this section. 

  1. page 14. The authors discuss the potency of fluvoxamiine as a Sigma 1 R receptor agonist, yet the literature on fluvoxamine's effect on seizures is mixed.  If fluvoxamine is a potent Sigma 1 R receptor agonist, why isn't it as effective for seizures as fenfluramine?  It could be a function of the biphasic dose response curve, but this requires further discussion and elaboration. 

Response

The reviewer raises an important point. We are actively investigating the answer to this question with the goal of better understanding the mechanism of action of fenfluramine for antiseizure efficacy. Although fenfluramine and fluvoxamine have not been directly compared experimentally, many different factors could account for this difference, including differences in experimental models, treatment protocols, dosages, pharmacological mechanisms of action at 5-HT and Sigma1R, and Sigma1R positive modulation as opposed to Sigma 1R agonism. These present some hypotheses for future testing. We have expanded Section 4.4 on Pages 13-14 to address these questions. 

Reviewer 2 Report

The manuscript by Martin et al, entitled "An Emerging Role for Sigma-1 Receptors in the Treatment of Developmental and Epileptic Encephalopathies" summarizes the latest advances in the investigation of the sigma 1 receptor and its possible relevance in the treatment of a given epileptic condition and other neuropathologies. While the revision is well structured and developed still important findings must be included in the text before publication.

Specific comments 

  1. Section 2. The authors should incorporate the existing literature demonstrating the localization of the sigma 1 receptor in the plasma membrane. It is important to mention that the sigma 1 receptor was originally described as an opioid receptor in the plasma membrane (Martin et al, 1976). Later, this receptor abandoned opioid classification being related to regulation of glutamate NMDA receptor, but always on the cell surface. This connection with the NMDA receptor, and other calcium channels, has been addressed in depth by several groups and should be reflected in the text. Since the glutamate is a particularly relevant transmitter in epilepsy, including data demonstrating the physical association between the sigma 1 receptor and the NMDAR to control the activation of the latter will reinforce the ideas presented in the manuscript. Membrane localization of sigma 1 receptors and its regulatory activity on a large number of GPC receptors is essential for the therapeutic efficacy of their ligands. Thus, drugs can reach the sigma 1 receptor without the need of internalizing into the neural cells, which has not been demonstrated in mature nervous system cells.
  2. Section 4.5. In the description of the pharmacological effects of sigma 1 receptor ligands it is mandatory to mention that the activity of these compounds is conditioned by those signalling proteins interacting with the receptor, biased activity of ligands. This concept has been demonstrated in a recent article through in vitro assays with recombinant proteins, in which sigma 1 ligands differentially alter the association of the sigma 1 receptor with calcium channels (NMDAR or TRPs) or with BiP.

Author Response

Reviewer #2:

The manuscript by Martin et al, entitled "An Emerging Role for Sigma-1 Receptors in the Treatment of Developmental and Epileptic Encephalopathies" summarizes the latest advances in the investigation of the sigma 1 receptor and its possible relevance in the treatment of a given epileptic condition and other neuropathologies. While the revision is well structured and developed still important findings must be included in the text before publication.

Specific comments 

  1. Section 2. The authors should incorporate the existing literature demonstrating the localization of the sigma 1 receptor in the plasma membrane. It is important to mention that the sigma 1 receptor was originally described as an opioid receptor in the plasma membrane (Martin et al, 1976). Later, this receptor abandoned opioid classification being related to regulation of glutamate NMDA receptor, but always on the cell surface. This connection with the NMDA receptor, and other calcium channels, has been addressed in depth by several groups and should be reflected in the text. Since the glutamate is a particularly relevant transmitter in epilepsy, including data demonstrating the physical association between the sigma 1 receptor and the NMDAR to control the activation of the latter will reinforce the ideas presented in the manuscript. Membrane localization of sigma 1 receptors and its regulatory activity on a large number of GPC receptors is essential for the therapeutic efficacy of their ligands. Thus, drugs can reach the sigma 1 receptor without the need of internalizing into the neural cells, which has not been demonstrated in mature nervous system cells.

 Response

The reviewer raises an important point regarding the localization of Sigma1R. Please see the expanded text on Pages 7-8, including reference to Martin et al. (1976) (Page 7, Line 125 [reference 58]). We have also illustrated the signaling components in the new Figure 2 on Pages 8-9 to show the physical association with Sigma1R and NMDAR.

  1. Section 4.5. In the description of the pharmacological effects of sigma 1 receptor ligands it is mandatory to mention that the activity of these compounds is conditioned by those signalling proteins interacting with the receptor, biased activity of ligands. This concept has been demonstrated in a recent article through in vitro assays with recombinant proteins, in which sigma 1 ligands differentially alter the association of the sigma 1 receptor with calcium channels (NMDAR or TRPs) or with BiP.

Response

We have added this information to the text on Page 8 and to the illustration in Figure 2 on Page 8. We have also added a reference (Cortes-Montero et al. 2019), which addresses ligand biases in interactions between Sigma1R and TRPs on Page 8 (Lines 150-155).

Additional Edits

In addition to these changes specified by the reviewers, we have made the following edits:

  1. We edited the statement that the Sigma1R was “cloned by Su in 1982” on Page 7 (Lines 124-126) to “cloned by Hanner et al. in 1996.”
  2. We added lines to the text and to Table 2 to indicate that the anticonvulsant phenytoin was historically the first identified Sigma1R positive modulator. These data provide strong evidence that Sigma1R positive modulators may be effective in epilepsy. Please see Page 12 (Lines 347-355) and Table 2 on Page 7.
  3. Geenen et al. (Reference 17) has been published and is now presented with the full citation in the reference list in place of “in press.” See also Supplemental Table 1 on Page 2.
  4. We have updated the text for clarity and readability in the Limitations and Conclusions sections (Pages 18-20, Lines 665-698; 726-736) and on Page 12, Lines 347-355.

All authors have participated in revising the manuscript. All authors have approved the revised manuscript as submitted. As edited, this manuscript does not use data similar to data provided in previous peer-reviewed publications, nor has this manuscript been posted on a preprint server.

Please do not hesitate to contact me if you require additional information related to our submission. Thank you for your consideration of our revised manuscript.

Sincerely,

Bradley S. Galer

Zogenix, Inc., 5959 Horton Street, Suite 500, Emeryville, CA, USA 94608

Phone: 484-675-5884; [email protected]